# Re-Evaluating the Role of PTHrP in Breast Cancer

**DOI:** 10.3390/cancers15102670

**Published:** 2023-05-09

**Authors:** Jeremy F. Kane, Rachelle W. Johnson

**Affiliations:** 1Program in Cancer Biology, Vanderbilt University, Nashville, TN 37232, USA; 2Vanderbilt Center for Bone Biology, Vanderbilt University Medical Center, Nashville, TN 37232, USA; 3Division of Clinical Pharmacology, Department of Medicine, Vanderbilt University Medical Center, Nashville, TN 37232, USA

**Keywords:** PTHrP, intracrine signaling, bone metastasis, breast cancer

## Abstract

**Simple Summary:**

Parathyroid-hormone-related protein (PTHrP) is produced in normal breast and breast cancer cells and causes bone destruction when expressed by cancer cells that have spread to the bone. When PTHrP is released, it binds to its receptor on neighboring bone cells; however, we now understand that PTHrP does not behave this way in breast cancer. Instead of binding to a receptor, PTHrP moves around inside the cells, which results in pro- or anti-tumor effects. This has caused confusion in the field since studies sometimes report conflicting results after deleting or increasing the expression of PTHrP. This review will improve our understanding of PTHrP and breast cancer by discussing the unique role PTHrP plays in breast cancer and how PTHrP signaling inside the cell and the different regions of PTHrP may play a role in the observed discrepancies.

**Abstract:**

Parathyroid-hormone-related protein (PTHrP) is a protein with a long history of association with bone metastatic cancers. The paracrine signaling of PTHrP through the parathyroid hormone receptor (PTHR1) facilitates tumor-induced bone destruction, and PTHrP is known as the primary driver of humoral hypercalcemia of malignancy. In addition to paracrine signaling, PTHrP is capable of intracrine signaling independent of PTHR1 binding, which is essential for cytokine-like functions in normal physiological conditions in a variety of tissue types. Pre-clinical and clinical studies evaluating the role of PTHrP in breast cancer have yielded contradictory conclusions, in some cases indicating the protein is tumor suppressive, and in other studies, pro-growth. This review discusses the possible molecular basis for the disharmonious prognostic indications of these studies and highlights the implications of the paracrine, intracrine, and nuclear functions of the protein. This review also examines the current understanding of the functional domains of PTHrP and re-evaluates their role in the unique context of the breast cancer environment. This review will expand on the current understanding of PTHrP by attempting to reconcile the functional domains of the protein with its intracrine signaling in cancer.

## 1. Introduction

### 1.1. A Brief History of Parathyroid Hormone-Related Protein (PTHrP) and the Receptor Binding Model

The parathyroid hormone-related protein (PTHrP), so named because of a similar N-terminal structure to parathyroid hormone (PTH), is alternatively spliced in humans as 139, 141, and 173 amino acid isoforms that are expressed at low levels by a variety of human tissues [1,2]. The sequence similarity to PTH allows PTHrP to bind to the PTH receptor (PTHR1, a G protein-coupled receptor) and induce downstream signaling cascades. Two main arms of signaling are triggered upon binding of PTHrP to PTHR1: one is cAMP (cyclic adenosine monophosphate)-driven and activates protein kinase A (PKA), and the other is phospholipase C (PLC)- and calcium (Ca^2+^)-driven and activates protein kinase C (PKC) [3]. PTHrP plays an important role in development but is well known for its role in humoral hypercalcemia of malignancy, where it was first described as a driving agent of the disease. Early clinical observations noted that patients with malignant cancers experience a dramatic increase in humoral calcium and weakening of bones, indicative of an increase in bone resorption [4]. This was similar to a disease known as hyperparathyroidism, where an increase in parathyroid hormone (PTH) stimulates osteoclastogenesis and bone resorption; however, it was noted that in these patients parathyroid dysfunction was not present, yet immunochemical analyses indicated that there were high levels of a PTH-like protein circulating through the bloodstream, and this “ectopic parathyroidism” was postulated to be secreted by the tumors themselves [5]. The circulating protein in question was later identified as a distinct protein with a region of high similarity to the PTH molecule and was purified from a number of cell lines, including a breast cancer cell line [6]. Thus, the “ectopic” PTH was identified as a distinct factor, PTHrP, and was subsequently studied as the key driver of hypercalcemia of malignancy.

### 1.2. Physiologic Functions of PTHrP

PTHrP is an essential gene for mouse neonatal survival and normal development of chondrocytes, which produce collagen. Complete genetic knockout of the PTHrP gene (parathyroid hormone-like hormone; *Pthlh*^−/−^) results in severely stunted chondrocyte proliferation, disrupted endochondral bone development, and asphyxia in newborn mice [7]. Inversely, an overexpression of PTHrP in murine embryonic chondrocytes causes a delay in the differentiation of chondrocytes and eventual dwarfism and an abnormally cartilaginous endochondral skeleton [8]. Interestingly, the delays in differentiation caused by chondrocyte overexpression of PTHrP subside after seven weeks and are otherwise normal, besides limb shortening. These observations highlight the importance of PTHrP in the initial progression of endochondral bones but its later dispensability in chondrocyte development. PTHrP expression continues to be endogenously expressed in adult mammals in osteoblast lineage cells as a mediator of bone turnover, where PTHrP binding to PTHR1 on osteoblasts induces osteoclast activation by inducing receptor activator of nuclear factor kappa-B ligand (RANKL) expression [9]. This leads to increased osteoclastogenesis and bone resorption, followed by osteoblast-mediated bone formation to restore the bone matrix, a tightly regulated process that when dysregulated leads to the hypercalcemia observed in metastatic cancers. PTHrP is also essential for lactation, and also occurs in a PTHrP-PTHR1-binding-dependent manner [10]. This is facilitated in a similar manner to bone resorption, where PTHrP binds to the PTHR1 on osteoblasts and induces osteoclast activation. In lactation, however, the production of PTHrP by breast epithelium produces circulating PTHrP, which mobilizes more bone calcium than can be replaced. This excess calcium is then sequestered for use in milk production [11].

However, PTHrP-PTHR1 binding is not the only mechanism of PTHrP activity. PTHrP has been shown to translocate to the nucleus immediately following translation, independent of PTHR1 binding. This mechanism is essential for the proliferation of vascular smooth muscle cells [12,13]. The action of PTHrP in these cells controls the cell cycle by degrading p27, though the precise interactions within the nucleus are still unclear [14]. These normal functions, divided loosely into the receptor-mediated paracrine activation of osteoclasts and intracrine nuclear cytokine-like activity, highlight the dynamic range of functions of PTHrP and suggest avenues that can be exploited by cancer. The intracrine signaling of PTHrP, particularly in cancer cells, is discussed in more detail later in this review.

### 1.3. PTHrP and the Vicious Cycle of Bone Destruction

Perhaps one of the better understood roles of PTHrP in cancer is its role in the so-called “vicious cycle of bone destruction” in bone metastases (Figure 1). High levels of PTHrP expressed by bone-metastatic tumor cells bind to PTHR1 on osteoblasts, inducing these cells to release RANKL, which in turn binds to RANK on immature osteoclasts and promotes osteoclastogenesis [15]. Osteoclasts then resorb the bone matrix, which is a reservoir for growth factors such as transforming growth factor beta (TGF-β), bone morphogenetic protein (BMP), and insulin-like growth factor (IGF) 1 and 2. These growth factors in turn stimulate the tumor to proliferate, which feeds the cycle forward [16]. Of particular note is the release of TGF-β, which in tumor cells triggers the SMAD (suppressor of mothers against decapentaplegic) cascade by binding to the TGF-β type II receptor. This forms a SMAD2/3/4 complex that translocates to the nucleus and activates the transcription factor Gli2, which increases transcription of pro-growth genes [17]. TGF-β and Gli2 are key regulators of PTHrP transcription, and their activation or inhibition greatly increase or decrease expression of PTHrP, further accelerating the cycle [18]. On a disease scale, this dysregulated bone resorption manifests as reduced bone integrity and greater fracture risk. In breast cancer patients, this translates to 60% of those with bone metastasis suffering from pathological fractures and a reduced median survival rate of two years [19]. This cycle remains the clinical focus of PTHrP’s role in metastatic cancers.

## 2. Clinical Significance of PTHrP in Breast Cancer: Complications in the Paracrine Model

### 2.1. PTHrP as a Prognostic Factor

Though the role of PTHrP in tumor-induced bone destruction is well supported by a consensus of data, the clinical data regarding the prognostic significance of PTHrP expression in breast cancer lacks the same harmony [20]. A retrospective study using immunohistochemical identification of PTHrP correlated its presence in 102 primary breast tumors with an overall negative outcome for the patient, independent of estrogen receptor (ER) status [21]. A similar study of 177 resected breast tumors correlated the expression of PTHrP with the progression of breast tumors, bone metastases formation, and a shorter overall patient survival [22]. Another clinical study determined that breast cancer bone metastases had a higher incidence of PTHrP expression than other sites of metastasis [23], though neither study differentiated between anatomic locations of the bone metastases. Nevertheless, these studies support the idea of a high-PTHrP-expressing tumor being more fit to colonize the bone and potentially incorporate the pro-proliferative functionality of PTHrP in the primary tumor. In contrast, a long-term (median of 67 months observation) prospective clinical study found that PTHrP expression in primary breast tumors correlated with a reduction in bone metastases and an overall better prognosis, while patients with primary tumors that were PTHrP-negative were more likely to develop-PTHrP-positive bone metastasis and an overall worse prognosis [24]. This observation was further validated and refined in a separate study, with the additional specification that reductions in nuclear PTHrP in primary tumors were associated with a poorer prognosis and that the PTHrP levels in aggressive breast tumors were lower than normal breast epithelial tissue, implying a protective role of nuclear PTHrP in primary breast tumors and patient outcomes [25]. Due to these conflicting data, PTHrP has seen limited use as a prognostic factor in overall breast cancer progression.

### 2.2. Inhibiting PTHrP in an Oncology Setting

Inhibition of PTHrP has been therapeutically explored in metastatic cancer, both to mitigate tumor-induced bone destruction as well as target the growth of the tumor itself. Pre-clinical mouse studies blocking PTHrP with small-molecule inhibitors and neutralizing antibodies have shown promise in reducing metastatic disease across many types of cancers, including colorectal cancer [26], melanoma [27], squamous cell lung carcinoma [28], and breast cancer [28,29,30], but most efforts have focused on targeting the upstream regulators of PTHrP, including TGF-β, GLI2, matrix metalloproteinases (MMPs), and p38 MAP kinases [31]. Although a clinical trial investigating PTHrP-neutralizing antibodies in breast cancer was initiated and stalled many years ago, no clinical trial inhibiting PTHrP has reached study completion. 

The bone destruction caused by metastatic breast cancer, on the other hand, is treated clinically by anti-resorptive therapies. Bisphosphonates, a class of drugs defined by two active phosphate groups, are currently used in the clinical setting to treat cancer-induced bone destruction. These drugs home to the bone and directly inhibit osteoclast activity and survival; in turn, they have been shown to improve the survival of postmenopausal women with metastatic breast cancer [32]. Another drug working on a similar principle, the RANKL inhibitor denosumab, has been clinically used as a downstream inhibitor of PTHrP-mediated bone disease [33,34]. Two studies with a primary endpoint of disease-free survival after denosumab treatment have been conducted, one evaluating denosumab as a single agent [33] and the other investigating the use of denosumab as an adjuvant therapy after conventional aromatase therapy [34]. Neither study yielded compelling evidence supporting denosumab as an agent to improve disease-free survival as a primary endpoint, though the adjuvant therapy did reduce the risk of clinical fractures associated with aromatase treatment. However, a secondary endpoint study of the adjuvant denosumab and aromatase inhibitor therapies suggests that targeting the PTHrP-driven pathway can delay metastatic disease in breast cancer patients and result in a significant though small (87.2% to 89%) increase in disease-free survival, although the study lacks the appropriate power to determine the therapeutic potential of denosumab [35]. Though inhibiting bone resorption has seen clinical success to some extent, the viability of direct modulation of PTHrP as a cancer therapy remains relatively unexplored in humans. Coupled with the poor efficacy of PTHrP as a prognostic indicator, it seems that a better understanding of how PTHrP acts in a breast cancer system at a molecular level is required to determine its clinical relevance.

## 3. Intracrine PTHrP in Cancer: Mounting Evidence

A potential cause of the contradictory clinical breast cancer data may be the simultaneous action of intracrine and paracrine signaling of PTHrP. An early study of MCF-7 cells, an estrogen/progesterone/glucocorticoid receptor-positive breast cancer cell line [36], showed that while they expressed the PTHR1, when they were stimulated with PTH they, puzzlingly, did not generate the cAMP signaling cascade expected following activation of the G protein-coupled receptor [37]. Other G protein-coupled receptors, such as those for calcitonin and prostaglandin E_2_, were able to induce cAMP production, suggesting that the lack of response to PTH stimulation was not due to a global inhibition of cAMP production in the MCF-7 cells but rather a detachment from PTHR1 and adenylyl cyclase. This phenomenon was investigated further in a study that evaluated the sensitivity of MCF-7 cells to PTH and PTHrP signaling through PTHR1, again finding that cAMP signaling was not activated following exposure to PTH or PTHrP. Furthermore, tetramethylrhomadine-tagged PTH was not detected at the cell surface or in intracellular endosomes, suggesting that not only was adenylyl cyclase not activated, but the PTHR1 was non-functional in the breast cancer cell line [38]. Interestingly, overexpression of PTHrP in MCF-7 cells downregulates tumor dormancy genes, including the tumor suppressor leukemia inhibitory factor receptor (LIFR) [39], in part through close association with the DNA in the LIFR promoter region [40]. This implies that an intracrine mechanism of PTHrP signaling may directly affect breast cancer cell outgrowth, and PTHrP-PTHR1 binding is un-involved (Figure 2). These findings do not seem to be limited to the MCF-7 cell line, as a recent study observed a similar mechanism in in vivo and ex vivo mouse mammary tumor cells derived from MMTV-PyMT mice [41]. In that study, the overexpression of PTHrP in PyMT tumor cells induced STAT5 activation and secretory differentiation, which was paired with an acceleration of mammary tumor development; however, these changes were not observed when PTHrP was added exogenously to the cell culture media. In addition, ablating PTHR1 in epithelial tissue and inhibiting the receptor with an antibody did not prevent the activation of STAT5 and development of secretory phenotypes or prevent the rapid formation of mammary tumors [41], providing further evidence that PTHrP intracrine signaling not only occurs in breast tumors, but appears to drive major cell growth programs. Taken as a whole, these studies shed light on a seemingly unique and understudied role of PTHrP that may have significant influence over breast tumor progression. Exactly how this is occurring, and a better understanding of the downstream effects of PTHrP intracrine signaling, have yet to be determined. 

## 4. Structural Insights: Reconciling the Intracrine and Paracrine Models

The key to understanding how intracrine PTHrP influences breast tumor growth and progression may lie in the structure of the protein itself. Human PTHrP is generated in three alternatively spliced isoforms of 139, 141, and 173 amino acids [42]; however, the instability of mRNA transcripts, as well as proteolytic cleavage, has obfuscated an understanding of the relative abundance of the isoforms in an in vivo environment. Complicating this matter further, mice are not capable of the same alternative splicing as humans, only generating a 139 amino acid isoform, limiting the scope of studying the abundance of PTHrP isoforms to human samples or in immune-compromised pre-clinical models [43]. Of the three human PTHrP isoforms, the 141 amino acid length isoform is most commonly associated with humoral hypercalcemia of malignancy and was first identified as containing a 36-amino-acid secretory signal and a 141 amino acid mature peptide sequence [44]. However, the human 139 amino acid isoform has been shown to increase breast cancer metastasis to bone in an induced murine expression study [42].

The regions of the mature peptide sequence have historically been roughly divided into four named functional regions (Figure 3), with the first being a PTHR1-binding region beginning at the N-terminus from amino acids 1–34, which is responsible for the multiplicity of functions through binding to PTHR1 [43], as described earlier in this review. The second is a relatively uncharacterized “midregion” from amino acids 35–66 that has been implicated in facilitating lung healing [45], and in combination with amino acids 67–94, is important for placental calcium transport in sheep [46]. The third is a nuclear localization signal (NLS) from amino acids 67–94, which is essential for the nuclear import of PTHrP in smooth muscle cells [13]. Following the NLS is an uncharacterized and historically unnamed “gap” region from amino acids 95–106. The fourth historically named domain is a C-terminal region from amino acid 107 to the end of the molecule, which has had a number of functions ascribed to it, including the osteostatin molecule from amino acids 107–111, which inhibits osteoclast-mediated bone resorption [47]. These regions have been studied as units with these general functions ascribed to them, though in many cases the exact delineation between regions has not been experimentally determined; rather, they exist as an artifact of restriction enzymes used in deletion studies. Of these historically named regions, the nuclear localization signal in particular has been examined for its role in facilitating the intracrine signaling of PTHrP.

### 4.1. Nuclear Localization of PTHrP

The nuclear trafficking of PTHrP is facilitated by importin-β in osteogenic sarcoma cells and is dependent on the presence of amino acids 66–94 of the PTHrP molecule [48]. This interaction with importin-β provided the original justification for naming the region the nuclear localization sequence, as well as its homology to an NLS present in human retroviral regulatory protein [49]. Interestingly, there is evidence that the region of amino acids 87–107 drives PTHrP nuclear and subsequent nucleolar localization in chondrocytes [50] and that this trafficking is important for conferring chondrocyte resistance to apoptotic death [51]. In addition, PTHrP nuclear entry is independent of PTHrP binding to PTHR1 chondrocytes and does not depend on endosomal trafficking that is typical of receptor-mediated nuclear translocation [52]. In a breast cancer model, the 67–101 amino acid region of PTHrP as a fragment was capable of nuclear localization and was able to induce the growth of MCF-7 and MDA-MB-231 cells [49]. Whether these variable sequences of importance to nuclear localization represent individual tissue-dependent nuclear localization signals or a bipartite nuclear signaling sequence, and whether intracrine PTHrP signaling requires these sequences, remains to be experimentally determined. The lack of clarity of the mechanism of PTHrP nuclear localization may be of importance to understanding PTHrP’s role in cancer. This may be particularly important when considering the conflicting human data on PTHrP’s prognostic meaning discussed earlier in this review, especially if PTHrP nuclear localization improves breast cancer patient survival [24,25].

### 4.2. The C-Terminal Region of PTHrP

The PTHrP C-terminal region, ranging from amino acid 107 to the end of the protein, has been determined by truncation experiments to play a variety of functions across different tissues. The aforementioned osteostatin pentapeptide is perhaps the most widely studied subregion of PTHrP due to its role in suppressing osteoclast-mediated resorption, contrary to the paracrine PTHrP function which promotes osteoclastogenesis. Osteostatin has been shown to have antimitogenic properties [53] and antioxidant functionality [54]. As a whole functional unit, the C-terminal region takes on additional functionality, most notably increasing the survival of human osteoblasts [55], as well as inducing their differentiation, by increasing the expression of vascular endothelial growth factor receptor 2 (VEGFR2) [56]. In contrast, the same region of PTHrP (amino acids 107–139) has been shown to exhibit an anti-proliferative effect on rat osteoblastic osteosarcoma cells, again independent of PTHR1 signaling [57], suggesting yet another dynamic function of intracrine PTHrP but this time through the C-terminus. The seemingly vital role of the PTHrP C-terminal region in modulating proliferation, combined with the lack of traditional PTHR1-mediated signaling in breast cancer highlights an unexplored potential role in PTHrP biology in cancer.

## 5. Proteolytic Processing of PTHrP: Re-Evaluating the Model in the Context of Breast Cancer

Perhaps the most significant case for a functional analysis of the domains and re-evaluation of the historical biological divisions of the PTHrP molecule is the widely reported propensity of PTHrP to undergo post-translational processing and the presence of truncated forms of the peptide in vivo. The processing of the N-terminal regions of the protein are well-documented in vitro, with furin in COS-7 cells [58] and pro-hormone thiol protease in lung cells [59]. N-terminal cleavage of purified PTHrP has also been observed by neprilysin [60] and prostate-specific antigen [61] in controlled test tube experiments. Bioinformatic techniques have also predicted with high certainty that prolyl oligopeptidase can cleave the N-terminal region of PTHrP [62] (Figure 3). Data providing evidence for processing of the NLS, “gap”, and C-terminal regions in vivo by these proteases have yet to be collected; however, the amino acid sequences of these regions include areas of high arginine and other basic amino acids, which are a favorable target for proteolytic cleavage by enzymes such as furin [63]. Furin itself is reported to be upregulated in some breast cancers and is associated with increased pro-survival calcium signaling [64] and worse overall patient outcomes [65]. Interestingly, fragments of PTHrP that were identified via immunoreactive assays to be amino acids 109–141 were found to be circulating in patients with a variety of cancers [66]. In addition, a portion of the PTHR1-binding domain and midregion (amino acids 12–48) was identified in patient plasma samples via mass spectrometry as a potential biomarker for metastatic breast cancer, implying that fragmented PTHrP circulates in a breast cancer disease state [67]. Taken together, these studies suggest that processing of PTHrP occurs in breast cancer and that this may disrupt or wholly remove the historically named functional regions of the PTHrP molecule. This processing may result in the formation of dynamic functional regions, highlighting the need to gain a better understanding of the consequences of these regions in breast cancer. 

## 6. Clinical Administration of PTH/PTHrP Analogs in Cancer Patients: Beneficial or Detrimental?

A synthetic molecule comprised of the first 34 amino acids of PTHrP, also known as abaloparatide, has been deployed as a therapy for osteoporosis, specifically in postmenopausal women. As an anabolic treatment, it has been shown to improve bone mineral density more successfully than the PTH analog, teriparatide [68], but is structurally similar to PTH since it lacks the PTHrP midregion, NLS, and C-terminus. Given the extent of bone loss caused by radiation, chemotherapy, and tumor-targeted therapies, there has been natural interest in the efficacy of PTH and PTHrP analogs in the oncology setting to mitigate low bone mass and prevent fractures in patients with bone metastatic disease. However, PTH and PTHrP analog use in patients with bone metastasis is limited due to a potential risk of osteosarcoma formation and the known role of PTHrP in metastasis-induced bone destruction [69]. In light of the apparent lack of PTHR1 functionality in breast cancer cells in preclinical models, a re-evaluation of the use of anabolic therapies in breast cancer patients may be timely. Breast cancer cell insensitivity to paracrine PTHrP signaling may allow abaloparatide to be administered in a premetastatic disease state to reduce osteoporotic bone loss, with no effect (either positive or negative) on tumor growth and progression.

Enticingly, intermittent treatment of a breast cancer mouse model with PTH (teriparatide) was shown to reduce the incidence of spontaneous bone metastasis to the hind limbs, with no change in extraskeletal metastases to the liver, lungs, or spleen when administered as a pretreatment or initiated 24 h after tumor inoculation [70]; however, a second study found an increase in breast cancer skeletal metastases to the forelimbs and ribs, but no change in dissemination to the hind limbs, following PTH pretreatment [71]. Studies using PTHrP analogs in preclinical bone metastasis models have not been published. The PTH studies suggest that PTH/PTHrP may have differential effects on tumor cell seeding in distinct skeletal niches, but this is only partially supported by the clinical data. PTH (teriparatide) and the PTHrP analog abaloparatide both significantly increase BMD in the lumbar spine, femoral neck, and hip [72,73], suggesting that the effect of PTH on the bone microenvironment may be similar across different skeletal sites. Interestingly, though, while PTH increases BMD in the radial shaft, it does not increase BMD in the distal radius [73], and in the second PTH preclinical study [71], this is where tumor burden was increased. Collectively, these data loosely suggest that PTH is protective against bone metastasis in the sites where it is most likely to stimulate BMD. Regardless, the preclinical data strongly suggest that any impact of PTH or PTHrP analogs on tumor progression, whether positive or negative, is likely to be mediated entirely through its paracrine effects on the bone microenvironment. This is supported by the data demonstrating that breast cancer cells are unresponsive to PTH or PTHrP treatment, and by the fact that PTH and abaloparatide lack the PTHrP midregion, NLS, and C-terminus, which appear critically important for PTHrP to directly regulate breast tumor progression. 

## 7. Generating a New Model: Unanswered Questions and Future Directions

The potential for fragmentary PTHrP peptides to not only exist in a breast cancer system, but also retain their individual functions as these peptides, presents a potential mechanism for the seemingly dynamic role of PTHrP observed in both clinical data and controlled preclinical experiments. Still, important questions remain. What specific amino acid residues contribute to the intracrine signaling of PTHrP in breast cancer? Does each region contribute equally, or are some more potent? To what extent does proteolytic cleavage of PTHrP occur in breast cancer cells, and are certain regions preferentially targeted? Does such cleavage produce functional peptides? Understanding precisely which regions of the protein contribute to pro-tumorigenic and tumor-suppressive phenotypes is important not only for answering these questions about the role of PTHrP in cancer progression but also for the successful use of the protein as a biomarker. This concept is not a new one and was ideated by Suva and colleagues [67], but it has remained stubbornly intractable due to a limited number of studies and resources directed toward detection and characterization of PTHrP fragments. In light of more recent preclinical data, perhaps moving beyond a simple PTHrP-low or PTHrP-high axis, to incorporating nuclear localization and the presence of functional regions, could create a more helpful PTHrP expression profile which better encapsulates the variety of effects of the protein through disease progression. This also could provide more information about whether the primary tumor or bone phenotype is of greater concern and whether drugs that bypass the intracrine signaling would improve patient outcomes. Further investigation into each of the functional regions, as well as the signals downstream of them, would be required to produce such a model.

## Figures and Tables

**Figure 1 cancers-15-02670-f001:**
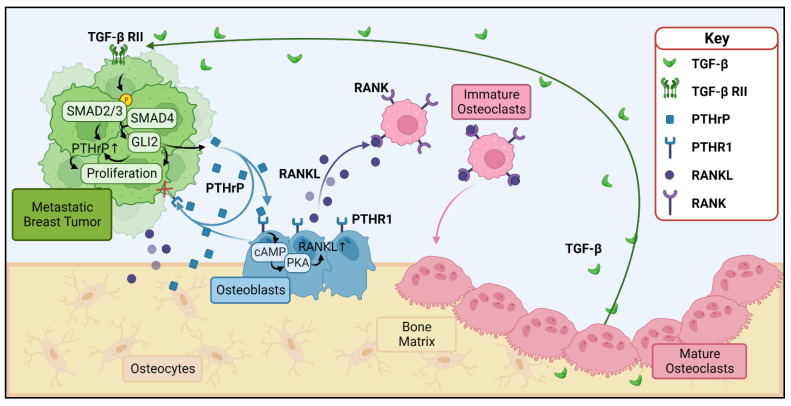
PTHrP and the vicious cycle of bone destruction. Metastatic tumors (green) and osteoblasts (blue) release PTHrP (blue squares), which binds to PTHR1 (blue receptors) to stimulate the release of RANKL (purple circles) from osteoblasts. Note that the osteocytes (tan) within the bone matrix secrete RANKL and PTHrP as well. RANKL binds to RANK (purple receptors) on osteoclast precursors (pink), inducing their maturation into osteoclasts and subsequent resorption of bone. Resorption of the bone matrix releases growth factors such as TGF-β, which in turn stimulate growth of the metastatic tumor and further release of PTHrP through activation of SMAD complexes and GLI2, which increase PTHrP transcription. PTHrP = parathyroid hormone related protein, PTHR1 = parathyroid hormone receptor 1, RANKL = receptor activator of nuclear factor kappa-B ligand, RANK = receptor activator of nuclear factor kappa B, TGF-β = transforming growth factor beta, TGF-β RII = transforming growth factor beta receptor type II. Figure generated with BioRender.com.

**Figure 2 cancers-15-02670-f002:**
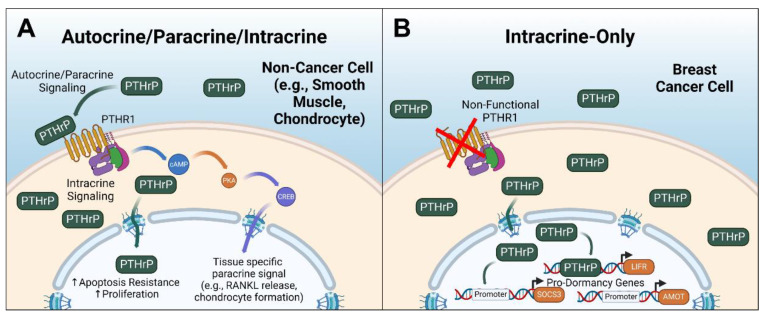
Overview of PTHrP paracrine and intracrine signaling. (**A**) Generalized signaling in non-breast-cancer cells, such as smooth muscle cells or chondrocytes, depicting the active PTHrP-PTHR1-mediated signaling and PTHrP intracrine signaling, which generally is pro-growth and prosurvival. (**B**) PTHrP signaling in breast cancer cells, which is exclusively intracrine and blocks the promoter regions of pro-dormancy genes. PKA = protein kinase A, CREB = cAMP response element-binding protein, SOCS3 = suppressor of cytokine signaling 3, AMOT = angiomotin. LIFR = leukemia inhibitory factor receptor. Figure generated with BioRender.com.

**Figure 3 cancers-15-02670-f003:**
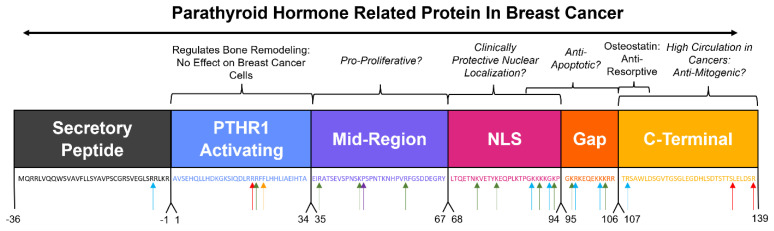
Contextualizing the historically named regions of PTHrP to breast cancer. Historic PTHrP biological domains and the corresponding sequence below in matching text color. Sequences potentially involved in important breast cancer progression pathways are italicized. Arrows indicate putative cleavage site by proteases that are upregulated in cancer (blue arrow = furin, red arrow = prostate-specific antigen, green arrow = pro-hormone thiol protease, yellow arrow = neprilysin, purple arrow = prolyl oligopeptidase family of serine proteases).

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
