# Peer review of "Re-Evaluating the Role of PTHrP in Breast Cancer"

_cancers, 2023, doi:10.3390/cancers15102670_

Round 1
Reviewer 1 Report
The authors’ lab has contributed to much of the data on the roles of Parathyoid hormone related protein ( PTHrP) in breast cancer. This review article discusses both tumor suppressor and tumor promoter roles of PTHrP reported in in breast cancer. The authors try to explain the conflicting roles on the basis of paracrine, intracrine and nuclear functions of the protein. The presentation is clear and the figures facilitate the interpretations.
While I congratulate the authors for an excellent review, I wish to point out a few concerns, addressing which will improve the article
(1) A recent comprehensive review article (2021) from the same corresponding author (reference 2) essentially covers many of the points as presented here. The authors should clearly spell out in what ways the present review is different and more informative.
(2) I suggest that the authors add a final section entitled “ Un-answered questions and future directions”
(3) The authors should add a list of abbreviations and their full nomenclature
Reviewer 2 Report
The authors present an interesting and well written review addressing the somewhat confusing and unresolved nature of PtHrP in metastatic bone disease. The review briefly introduces the signaling mechanisms of PTHrP and explains in function in bone under normal physiologic conditions. Ultimately the authors’ highlight how in some cases it can support or restrict tumor in bone. The authors go on to discuss how the somewhat contradictory results still need further investigation to understand a potential therapeutic intervention. The authors mainly focus on breast cancer in bone but many other cancers are likely to have similarities such as kidney, bladder and prostate.
Prior to publication I would suggest some considerations for the authors to enhance the information conveyed to their targeted audience. The greatest concern I have is that the synthesis of sections 2 (therapeutics) and section 4 (structural/function of PTHrP) are not fully synthesized in the final section 5. Specifically how would the authors advocate for distinct functional structure domains of PTHrP to be used with metastatic breast cancer patients? Anabolic treatments have largely been shunned in tumor induced bone disease that could be discussed further and why we still remain in anti-resorptive as opposed to osteogenic or anabolic treatment for bone loss could be more explicitly discussed.
Minor comments and suggestions:
1) I’m surprised at the lack of TYMLOS/Abaloparatide in either the first or second sections, but also as a fragment PTHrP related peptide this seems to need a little more insight. Would the authors propose more specific mechanisms based on section 3 to use Abaloparatide in acute or distinct common treatment such as radiotherapy or surgeries?
2) Does the anatomic location of the metastases matter for PTHrP? Are all bone created equally?
3) Since the review focuses on breast cancer and osteolytic bone disease is there a specific FRAX score/menopause endocrine interplay that could be important determination of PTHrP utility in patients?
4) The figure makes it appear that no bone matrix is being lost. Also in the figure it appears that only tumor cells secrete PTHrP and only OC’s have the receptor?
5) Would distinct genetic or pathological subtypes of breast cancer in bone have distinct responses to PTHrP signaling? For example do women with ER+ breast cancer have different survival benefits for PTHRP expression than say TNBC?
6) Is it possible to diagnose intracellular localization of PTHrP as a diagnostic in clinical breast cancer in bone decalcified biopsies?
